# Assessing the effects of an 8-week mindfulness training program on neural oscillations and self-reports during meditation practice

**Julio Rodriguez-Larios**[1]*, **Kian Foong Wong**[2], **Julian Lim**[2]

**1** Brunel University London, London, United Kingdom, **2** Centre for Sleep and Cognition, Yong Loo Lin School of Medicine, National University of Singapore, Singapore, Singapore

* julio.rodriguezlarios@brunel.ac.uk

**Data Availability Statement:** All raw data and MATLAB scripts are publicly available through the Open Science Framework repository (see https://osf.io/r879/).

## Abstract

Previous literature suggests that mindfulness meditation can have positive effects on mental health, however, its mechanisms of action are still unclear. In this pre-registered study, we investigate the effects of mindfulness training on lapses of attention (and their associated neural correlates) during meditation practice. For this purpose, we recorded Electroencephalogram (EEG) during meditation practice before and after 8 weeks of mindfulness training (or waitlist) in 41 participants (21 treatment and 20 controls). In order to detect lapses of attention and characterize their EEG correlates, we interrupted participants during meditation to report their level of focus and drowsiness. First, we show that self-reported lapses of attention during meditation practice were associated to an increased occurrence of theta oscillations (3–6 Hz), which were slower in frequency and more spatially widespread than theta oscillations occurring during focused attention states. Then, we show that mindfulness training did not reduce the occurrence of lapses of attention nor their associated EEG correlate (i.e. theta oscillations) during meditation. Instead, we find that mindfulness training was associated with a significant slowing of alpha oscillations in frontal electrodes during meditation. Crucially, frontal alpha slowing during meditation practice has been reported in experienced meditators and is thought to reflect relative decreases in arousal levels. Together, our findings provide insights into the EEG correlates of mindfulness meditation, which could have important implications for the identification of its mechanisms of action and/or the development of neuromodulation protocols aimed at facilitating meditation practice.

## Introduction

Mindfulness is a type of meditation practice that consists of paying attention to the present moment non-judgmentally [1]. This is usually cultivated through focused meditation, in which a particular object (e.g. the breath) is chosen as the target of attention [2, 3]. The practice of mindfulness has become popular in western cultures in the last years due to its putative

**Funding:** This study was funded by the start-up funding from the National University of Singapore to Julian Lim.

**Competing interests:** The authors have declared that no competing interests exist.

health benefits [4]. The mechanisms of action behind mindfulness are still debated and they are likely to involve a wide variety of factors [5, 6].

It has been proposed that some of the positive effects of mindfulness on mental health could be mediated by reductions in mind wandering [7], which can be defined as the emergence of spontaneous, self-generated thoughts that often entail memories, future plans or fantasies [8]. In support of this idea, previous studies have shown that mindfulness trait is negatively correlated to self-reported mind wandering [9] and that mindfulness training reduces mind wandering during different cognitive tasks [10, 11]. In addition, excessive mind wandering has been associated to poor mental health [12], probably due to its link to rumination and worry [13–15].

The neural correlates of meditation and mind wandering have been investigated through Electroencephalography (EEG). EEG is a non-invasive method that allows to record synchronized activity of large populations of neurons that are arranged orthogonal to the scalp [16]. The EEG signal is dominated by oscillatory electrical activity that is normally referred as neural oscillations. Neural oscillations have been classified according to their peak frequency (e.g. alpha = 8–13 Hz; theta = 4–8 Hz, etc.) and their occurrence has been associated to different cognitive functions and mental states [17, 18].

The EEG correlates of meditation depend on the type of meditation practice, the level of expertise of the participants and the 'control' condition used as a baseline [19]. In this way, it has been shown that breath focus meditation (relative to rest) is associated to decreases in alpha/beta amplitude and individual alpha frequency in experienced practitioners [20–22]. Because alpha power has been positively associated to mind wandering [23], its decrease during meditation (relative to rest) is thought to reflect reduced mind wandering [20]. This idea is further supported by studies demonstrating significantly reduced mind wandering during meditation in experienced meditators relative to novices [20, 24].

In this study, we assessed whether mind wandering during meditation (and their associated neural correlates) change significantly in novices after 8 weeks of mindfulness training. For that purpose, we recorded EEG during meditation before and after an 8-week mindfulness training (21 active and 20 waitlist controls). Using an experience sampling paradigm, we prompted participants during meditation practice to report their level of mind wandering and drowsiness. Our (pre-registered) hypothesis was that mind wandering during meditation practice would be reduced after mindfulness training and that this would be reflected in EEG neural oscillations. To test this hypothesis, we first characterized EEG modulations associated with mind wandering during meditation. Then, we assessed whether mind wandering and/or the EEG correlates of meditation practice changed significantly after mindfulness training.

## Methods

### Participants

48 participants (28 females) were recruited for the study. To be eligible for the study, participants had to be a National University of Singapore student between 21 and 35 years old. In addition, participants had to report moderate to high levels of perceived stress (Perceived Stress Scale score > 14), be willing to participate in an 8-weeks mindfulness course and be meditation-naïve. Exclusion criteria were: i) chronic physical or psychiatric illness, including all major Axis I and II disorders, ii) history of drug or alcohol abuse, and iii) long-term medication use. 35 participants were allocated to treatment (i.e. mindfulness training) and 24 to waitlist. The average age was 23.81 (SD = 2.59). EEG was recorded before and after 8 weeks in 21 treatment and 20 control participants. The discrepancy between recruited participants and the actual sample size was due to dropouts. In this regard, note that the study was run during

the COVID-19 pandemic, which significantly hindered assistance to the programme and data collection sessions.

The study was conducted at the National University of Singapore (NUS). The study was approved by NUS Intuitional Review Board and conducted in accordance to the 1965 Helsinki declaration and its later amendments. Written informed consent was obtained from all participants and participants were reimbursed with money for their time. Participants were recruited and data was collected between 3 of December 2020 and 3 of December 2022.

## Mindfulness training

We used a program modelled on the standard Mindfulness-based stress reduction (MBSR) program developed for adults by Kabat-Zinn [25]. The contents of this training included becoming aware of one's attention, intention, and attitude, how to conduct daily activities (e.g. eating and conversation), and how to use mindfulness strategies in particular stressful situations. Participants attended weekly 2.5 hour group sessions for eight weeks with a full-day mindfulness retreat between week 6 and week 7, all led by an instructor experienced in delivering mindfulness-based interventions. Sessions were face-to-face in a group setting or via video conferencing. For the post-pre training comparison we only included participants that attended at least 5 sessions.

## Design and task

Participants went through 3 experimental conditions while EEG was recorded. These conditions were: non-meditative rest (5 min), uninterrupted meditation (10 min) and interrupted meditation (~25 minutes). In the interrupted meditation condition, participants were presented with a bell sound at random time intervals between 30 and 90 seconds and asked to report their level of focus (from completely mind wandering to completely focused on the breath) and drowsiness (from very alert to falling asleep) on a 5-point scale using the keyboard (see **Fig 1A**). The specific instructions presented to the participants during the meditation conditions were:

- Uninterrupted meditation: *'In this part of the experiment, you will be asked to do a breath focus meditation with your eyes closed. We ask you to pay attention to wherever you feel the breath most clearly—either at the nostrils, or in the rising and falling sensation of your abdomen. There is no need to control your breath, just let it come and go naturally. Every time your mind wanders in thought (which is completely normal) gently return it to the sensation of breathing'*

- Interrupted meditation: *'In this part of the experiment we will also ask you to perform a breath focus meditation with your eyes closed. However, this time you will be interrupted several times throughout your meditation practice. You will hear a sound after which you will have to open your eyes and answer two questions. Then you will be asked to close your eyes and go back to the sensation of breathing. This part of the experiment has a total duration of 25 minutes'*

Note that in the interrupted meditation condition, only the 5 seconds before the probe were used for EEG analysis for consistency with previous studies [20, 26]. These EEG epochs were sorted into the categories 'mind wandering', 'focused attention', 'drowsiness', or 'alert' depending on self-reports. Each category was composed of epochs in which participants report one the two extreme values on the scale. For example, the category 'mind wandering' included epochs in which participants reported either 'somewhat mind wandering' (point 2 in the *mind wandering—focus scale*) or 'completely mind wandering' (point 1 in the *mind wandering—*

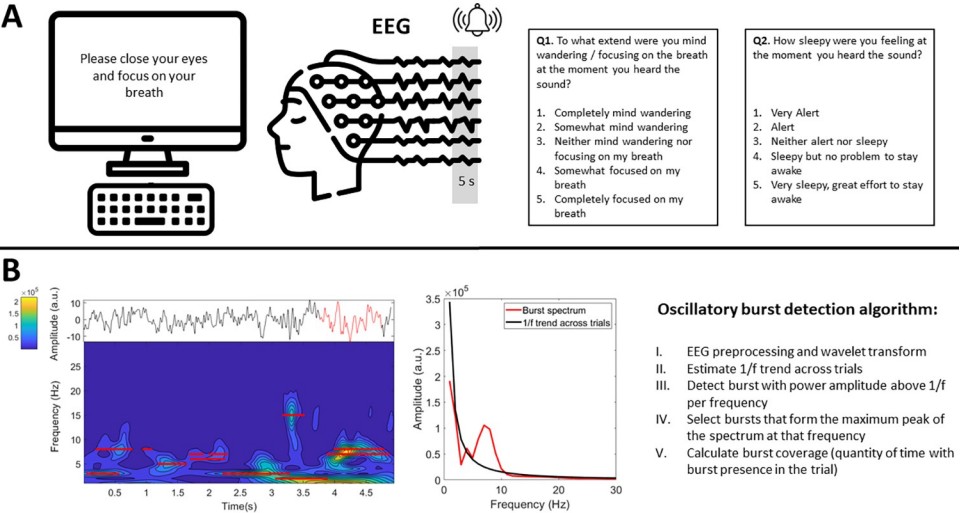

**Fig 1. A)** Experience sampling paradigm. Participants were interrupted during meditation practice with a bell sound to report their level of mind wandering and drowsiness while EEG was recorded. The last 5 seconds of the EEG signal before the bell sound were used for later analysis. **B)** Depiction of oscillatory bursts detection algorithm. The left panel depicts the raw EEG signal (top) and the time-frequency representation (bottom) of one trial. The right panel depicts the power spectrum of an exemplary oscillatory burst around 7 Hz (corresponding to the raw EEG signal marked in red in the left panel).

*focus scale*). Hence, in order to compare mind wandering vs breath focus and drowsy vs alert we did not use epochs in which participants reported 3 on the scale (i.e. *'neither mind wandering or focusing'*, *'neither alert or sleepy'*).

## EEG recordings

EEG data were recorded using a BrainProducts MR+ amplifier with a 64-channel actiCAP (standard 10–20 electrode positioning). Sampling rate was 250 Hz and the reference and ground electrodes were FCz and Fpz respectively. All electrode impedances were brought below 10 kΩ before the start of the recording.

## EEG analysis

EEG analysis was performed in MATLAB R2023a using custom scripts, EEGLAB (Delorme & Makeig, 2004) and Fieldtrip functions [27].

Data cleaning was performed using an automatic preprocessing pipeline based on EEGLAB functions. First, data were re-referenced to common average (function *pop_reref*) and filtered between 1 and 30 Hz (function *pop_eegfiltnew*). Abrupt noise in the data was removed using the Artifact Subspace Reconstruction method (function *clean_asr* with a cut-off value of 20 SD; see [28]. Noisy electrodes were detected automatically (function *clean_channels*) with a threshold of 0.5 and later interpolated (function *pop_interp*). Independent component analysis (ICA) (function *pop_runica*) and an automatic component rejection algorithm [29] were employed to discard components associated with muscle activity, eye movements, heart activity or channel noise (threshold = 0.7).

In order to detect oscillatory activity in the EEG signal, a recently developed algorithm was used [30]. In short, EEG data was first transformed to the time-frequency domain using 6-cycles Morlet wavelets as implemented in the MATLAB function *BOSC_tf* [31] between 1 and 30 Hz with a frequency resolution of 1 Hz. Then an estimate of the amplitude of the

aperiodic 1/f trend was obtained by fitting a straight line in log–log space to the average EEG frequency spectrum per electrode after excluding frequencies forming the maximum peak [32–34]. Oscillatory bursts were defined as time points in which the amplitude at a specific frequency exceeded the estimate of aperiodic activity for at least one full cycle (e.g. 100 ms for a 10 Hz oscillation). In order to rule out the possibility that the detected oscillatory bursts were artifactually originated from aperiodic activity or from non-sinusoidal properties of a different rhythm, only oscillatory bursts that formed the peak with the greatest prominence of the 1/f-subtracted frequency spectrum were selected. Using this algorithm, we obtained the quantity of time (seconds) in which oscillatory activity was detected (i.e. burst coverage) in each frequency (1–30 Hz), electrode, subject and experimental condition. See **Fig 1B** for a depiction of the analysis pipeline.

Once oscillatory bursts were identified, we calculated the individual alpha frequency through its centre of gravity [35], which can be defined as:

$$\frac{\sum_{i=1}^{n} fi * bci}{\sum_{i=1}^{n} bci}$$

where *fi* is the frequency, n is the number of frequency bins between 7 and 14 Hz, and *bci* the burst coverage for each frequency *fi*.

## Statistical analysis

For the behavioural analysis, two one-way repeated-measures ANOVA were performed with the JASP software [36]. The between subject factor was session (post vs pre) and the within subject factor was level of drowsiness or level of mind wandering.

For the EEG data, a cluster-based permutation statistical test (see Maris & Oostenveld, 2007) was used to assess the statistical significance of condition-related differences in oscillatory burst coverage. In short, this type of test controls for the type I error rate arising from multiple comparisons using a non-parametric Montecarlo randomization and taking into account the dependency of the data by the formation of clusters (in neighbouring electrodes and/or frequencies). Paired-samples t-test was chosen as the first-level statistic to compare oscillatory burst coverage between experimental conditions (i.e. mind wandering vs breath focus, drowsy vs alert). Independent-samples t-test was chosen as the first-level statistic for the comparison of the change in burst coverage (post–pre training) between groups (treatment vs controls) during the meditation conditions (interrupted and uninterrupted). Effect size of significant clusters was estimated using Cohen's *d* statistic, which is calculated by dividing the mean difference between conditions by their pooled standard deviation [38].

## Results

### Oscillatory correlates of lapses of attention due to mind wandering and drowsiness

We first assessed the oscillatory correlates of attentional lapses (due to mind wandering or drowsiness) during the interrupted meditation condition. Permutation tests revealed that both mind wandering and drowsiness were associated with an increased occurrence of delta/theta oscillations (~3–6 Hz) as quantified through oscillatory burst coverage (see **Fig 2**). For the mind wandering effect, a significant positive cluster with a posterior distribution and a frequency span of 3–6 Hz was found (*tcluster* = 78.38; *pcluster* = 0.0079; *d* = 0.48). For the drowsiness effect, a significant positive cluster involving the majority of electrodes and a frequency

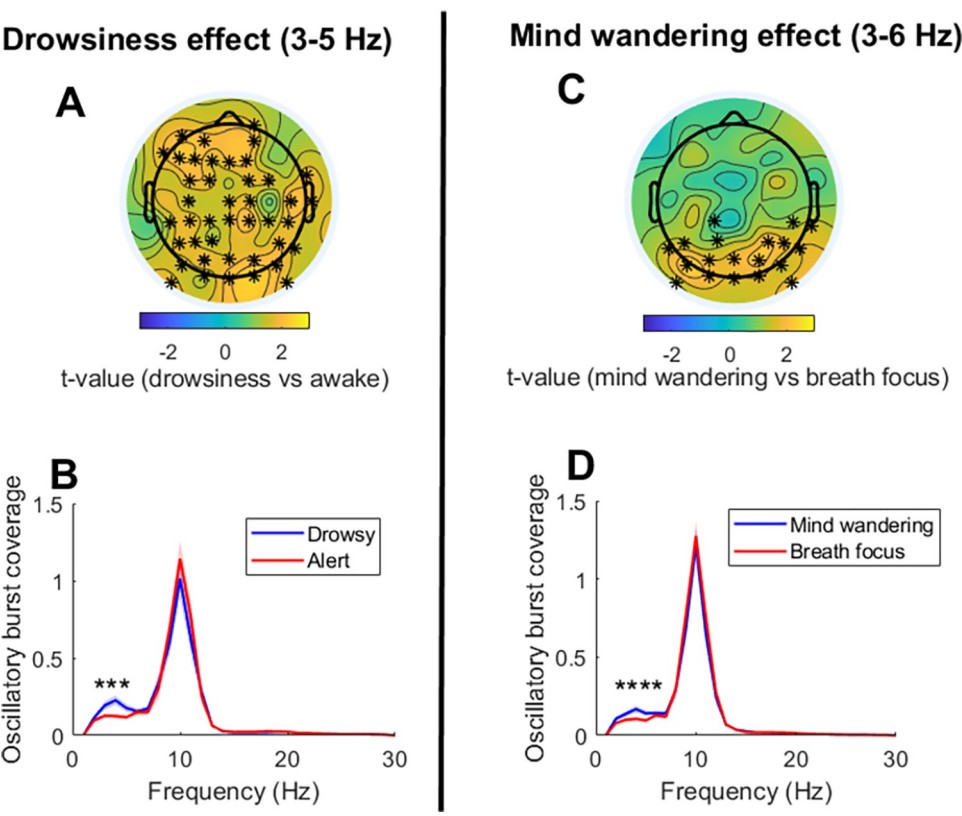

**Fig 2. Oscillatory correlates of mind wandering and drowsiness during meditation practice.** The upper panels (**A** and **C**) depict the topographical distribution of t-values for the drowsiness and mind wandering effects in the delta/theta range (3–6 Hz). Electrodes forming significant clusters at p<0.025 are marked with asterisks. The lower panels (**B** and **D**) depict the mean oscillatory burst coverage (with standard error in shade) of the identified significant clusters for each condition and frequency. Frequencies showing significant differences are marked with asterisks.

span of 3–5 Hz was found ($tcluster$ = 148.20; $pcluster$ = 0.004; d = 0.53). No significant mind wandering or drowsiness effects were revealed for individual alpha peak frequency.

## Characterizing theta oscillations during attentional lapses and focused attention states

Theta oscillations have been previously associated with both drowsiness and focused attention, a phenomenon that has been called *'the theta paradox'* [39]. Based on our results and previous literature, we performed exploratory analysis to further characterize theta oscillations during lapses of attention relative to focused attention states. For this purpose, we compared the spatio-spectral characteristics of theta oscillations during focused attention, drowsy and mind wandering trials within subjects and across sessions. Specifically, we calculated the center of gravity of theta oscillations in space (anterior-posterior and left-right dimensions) and frequency (4–7 Hz).

Our results show that theta oscillations during focused attention states had a more frontal distribution than theta oscillations during drowsiness ($t(74)$ = 2.47; $p$ = 0.015; $d$ = 0.28) and mind wandering states ($t(74)$ = 2.22; $p$ = 0.028; $d$ = 0.24). In addition, the peak frequency of theta oscillations for focused attention states was quicker relative to drowsiness ($t(74)$ = 3.57; $p$ = 0.0017; $d$ = 0.74), and mind wandering ($t(74)$ = 2.75; $p$ = 0.010; $d$ = 0.51). No significant

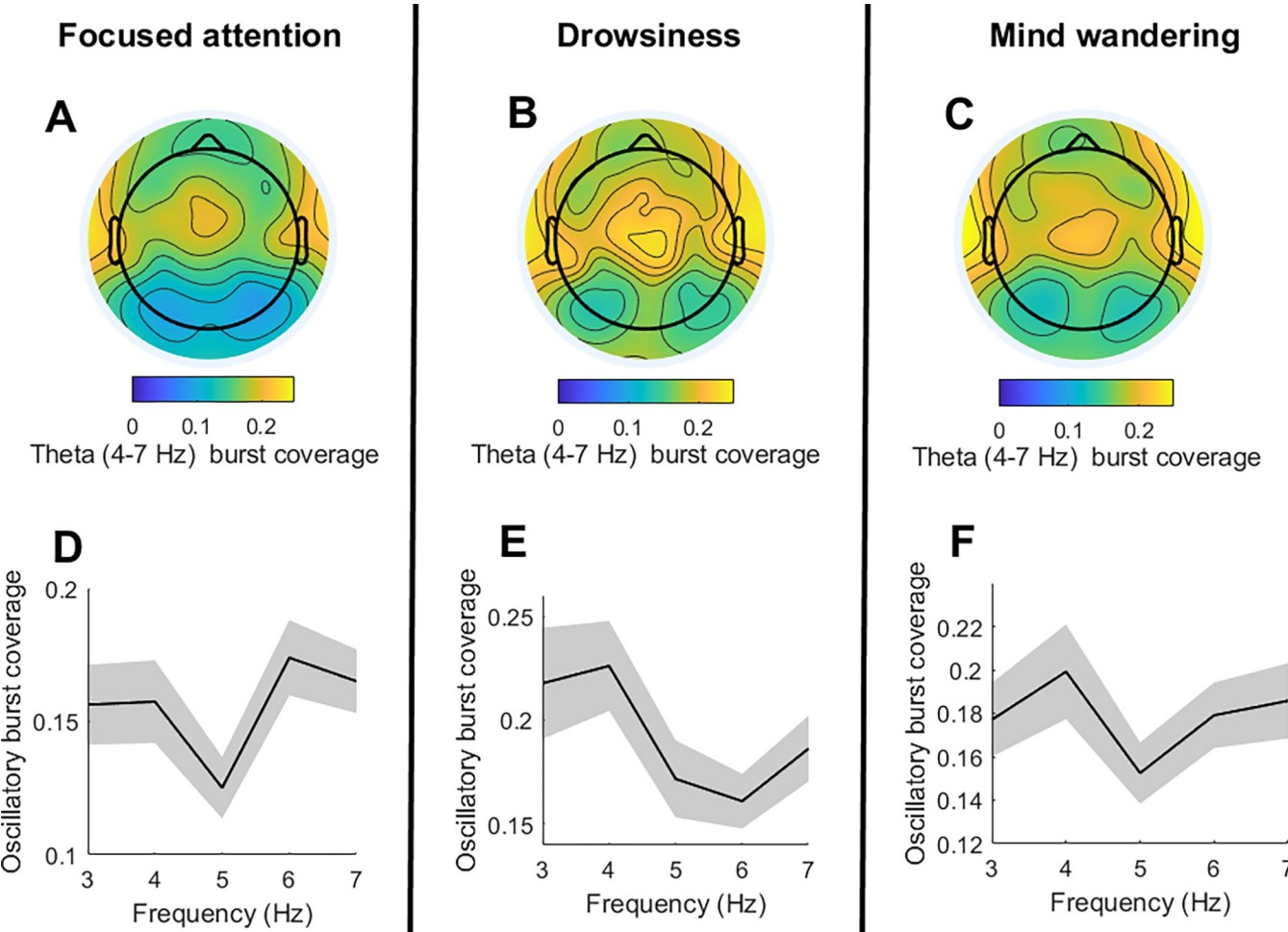

**Fig 3. Spatio-spectral characteristics of theta oscillations during focused attention, drowsiness and mind wandering in the context of meditation practice. A-C)** The upper panels depict the topographical distribution of Theta (4–7 Hz) oscillatory bursts coverage (i.e. time with theta oscillations presence) across subjects and sessions per condition. **D-F)** The lower panels depict mean oscillatory bursts per frequency and across electrodes, subjects and sessions (shades mark standard error).

differences were found between the spatio-spectral characteristics of theta oscillations associated to mind wandering and drowsiness.

Together, our exploratory analysis revealed that theta oscillations during lapses of attention (due to mind wandering or drowsiness) are slower and more posterior than theta oscillations occurring during focused attention states. For the visualization of these results, the mean topography and theta (4–7 Hz) frequency distribution of focused attention, mind wandering and drowsy trials (across subjects and sessions) are depicted in **Fig 3**.

## Experience sampling in treatment and control groups

Our pre-registered hypothesis was that lapses of attention during meditation practice due to mind wandering would be reduced after mindfulness training (see https://doi.org/10.17605/OSF.IO/DQ94G). If this was the case, we would expect a significant *Group*Session* interaction in the average score of the *mind wandering—focus* scale thereby reflecting a significant change in the treatment but not in the control group. Contrary to our hypothesis, Repeated Measures ANOVA did not reveal a significant *Group*Session* interaction in the *mind wandering—*

**Table 1. Mean and standard deviation of 5-point experience sampling scales for each group (treatment and control) and session (pre and post).**

| Scale | Treatment Pre | Treatment Post | Control Pre | Control Post |
|---|---|---|---|---|
| Mind wandering–focused | 3.35 | 3.26 | 3.05 | 3.03 |
| | (SD = 0.74) | (SD = 0.91) | (SD = 0.6) | (SD 0.79) |
| Alert–drowsy | 2.87 | 2.56 | 2.85 | 2.55 |
| | (SD = 0.92) | (SD = 0.93) | (SD = 0.85) | (SD 0.7) |

*focused* scale ($F(1,40) = 0.15$, $p = 0.69$) nor in the *alert-drowsy* scale ($F(1,40) = 0.05$, $p = 0.94$). The mean scores (and their standard deviation) for each scale per group and session are depicted in Table 1. In this regard, both treatment and control groups show a similar decrease in mind wandering and drowsiness in the second session. This decrease was significant for drowsiness ($F(1,40) = 7.50$, $p = 0.009$) but not for mind wandering ($F(1,40) = 0.3$, $p = 0.58$).

In addition, we assessed whether, in line with previous reports [24, 26], the mean level of focus was anti-correlated to the mean level of drowsiness. In fact, Pearson correlation revealed a negative correlation between average scores of the *mind wandering–focused* scale and the average scores of the *alert–drowsy* scale ($r = -0.31$; $p = 0.0014$) across subjects and sessions.

## Changes in neural oscillations after mindfulness training

In order to assess potential modulations in neural oscillations associated with mindfulness training, we compared oscillatory bursts changes (post–pre training) between treatment and control groups during both the interrupted and the uninterrupted meditation conditions.

No significant cluster was identified for the interrupted meditation condition for either burst coverage (at each frequency) or individual alpha frequency. For the uninterrupted meditation condition, a significant frontal cluster was identified for individual alpha frequency ($t_{cluster} = -77.11$; $p_{cluster} = 0.005$; $d = 1.05$; **Fig 4A–4C**) and no significant clusters were identified when assessing burst coverage at each frequency. Thus, these results show that alpha oscillations in frontal electrodes were significantly slower in the second (uninterrupted) meditation session for the treatment group but not for the control group.

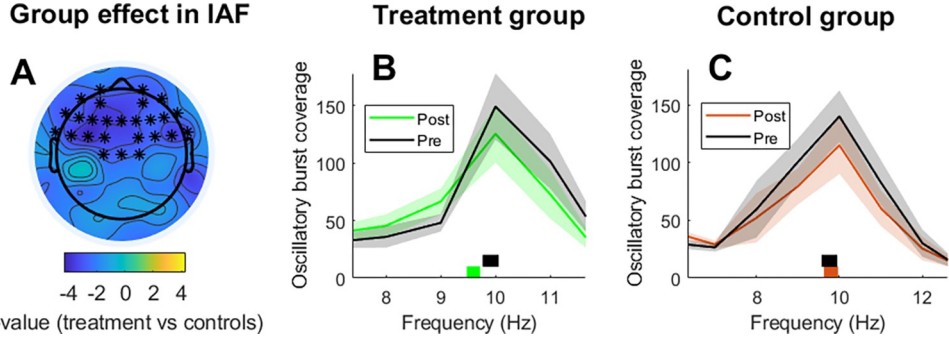

**Fig 4. Changes in individual alpha frequency associated with mindfulness training. A)** The left panel depicts the topographical distribution of t-values for the Group effect, which consists on the group comparison (treatment vs controls) of alpha frequency changes (post–pre) during meditation. Electrodes forming significant clusters at p<0.025 are marked with asterisks. **B-C)** The middle and right panels depict oscillatory burst coverage in the alpha range and individual alpha frequency per session for the treatment and control groups respectively. Individual alpha frequency was estimated as the center of gravity between 7 and 14 Hz and it is depicted in the lower part of each panel with rectangles (rectangle's width represents standard error).

## Discussion

This study investigated the EEG correlates of lapses of attention during meditation practice in order to assess their (hypothesized) reduction after mindfulness training. Lapses of attention during meditation (due to drowsiness or mind wandering) were associated to an increased occurrence of theta oscillations. Further analysis revealed that theta oscillations occurring during attentional lapses were slower in frequency and more widespread than those occurring during focused attention. Contrary to our hypothesis, our results show that neither lapses of attention nor the here identified EEG correlates (i.e. emergence of slow and widespread theta oscillations) were significantly reduced after mindfulness training. Instead, we found that mindfulness training was significantly associated to a slowing of frontal alpha oscillations during meditation practice.

### Oscillatory correlates of mind wandering and drowsiness

Our results revealed a greater prominence of delta/theta oscillations during lapses of attention in the context of meditation practice, which is line with previous studies [26, 40, 41]. However, this finding is not in line with a recent review showing that lapses of attention have been more consistently associated with relative increases in alpha power (8–14 Hz) during different cognitive tasks [23]. In fact, in a previous report, we showed relative increases in alpha (but not theta) power during mind wandering in the context of meditation practice [20]. Based on our results and previous literature, we speculate that mind wandering with drowsiness is associated with a relative increase in the occurrence of theta oscillations while mind wandering without drowsiness is associated with a relative increase in the occurrence of alpha oscillations. In support of this idea, we have previously reported that the theta power increase associated with mind wandering is positively correlated to drowsiness levels [26]. In the same line, we show in the current study that self-reported mind wandering and drowsiness levels were positively correlated and that their neural correlates were qualitatively similar (i.e. spatially widespread low-theta oscillations). However, direct evidence to confirm the relationship between alpha / theta oscillations and mind wandering / drowsiness is lacking and more research is needed. For that purpose, an interesting possibility would be to train participants in the identification of phenomenological categories so they can better characterize their attentional lapses in experience sampling paradigms [42, 43].

### The theta paradox during meditation practice

In addition to our pre-registered analysis, we performed exploratory analysis to characterize theta oscillations during meditation practice. These analyses were motivated by the so-called *'Theta Paradox'* which refers to the apparently contradictory emergence of theta oscillations during both drowsiness and focused attention states [39]. Our results show that theta oscillations during lapses of attention have a slower frequency (~4 Hz) and a more posterior distribution than theta oscillations occurring during focused attention states (which were centred around 6 Hz in midfrontal electrodes). Based on previous literature, our interpretation is that the ~ 6 Hz midfrontal theta typically observed during focused attention states [44] reflects selective cortical inhibition of a default mode network node [45] while the widespread (and slower) theta observed during drowsiness would reflect a more general cortical inhibition that would include task-relevant areas [46, 47]. Thus, this exploratory analysis suggests that the frequency and spatial specificity of theta oscillations might determine their phenomenological counterpart. Therefore, our results could have important implications for the development of neuromodulation protocols aimed at enhancing theta oscillations to promote focused attention states [48] or sleep pressure [49, 50].

### The effect of mindfulness training on attentional lapses

Mindfulness training has been associated with a reduced number of lapses of attention, as quantified through behavioural performance in different cognitive tasks and questionnaires [51]. However, no previous study investigated the effect of mindfulness training on lapses of attention due to mind wandering during meditation practice through experience sampling. Based on previous studies showing reduced mind wandering during meditation in experienced meditators relative to novices [20, 24], we hypothesized that 8-weeks of mindfulness training would lead to reduced number of lapses of attention due to mind wandering during meditation practice. Contrary to our pre-registered hypothesis, we find no differences in self-reported mind wandering levels after meditation training.

There are at least two possible explanations for the lack of changes in mind wandering during meditation after mindfulness training. One possibility is that 8 weeks of mindfulness training reduces mind wandering during cognitive tasks [51] but not during meditation. Another possibility is that reduced mind wandering during meditation after mindfulness training only occurs when tested in meditation practices that last longer. In this way, the interrupted meditation condition in this study only lasted 25 minutes while the meditation condition of studies that reported differences between experienced meditators and novices in mind wandering lasted around 60 minutes [20, 24]. Therefore, future studies combining longer meditation sessions with comprehensive cognitive testing are needed to elucidate under which conditions mindfulness training reduces mind wandering.

### On alpha slowing during meditation practice

We show that the frequency of alpha oscillations during the second uninterrupted meditation session was significantly slower in participants that followed the 8-weeks Mindfulness training (relative to controls). This result is in line with previous studies showing a slowing of individual alpha frequency or a power increase in the lower alpha band (i.e. 7–10 Hz) during meditation (relative to a control condition) in experienced practitioners [20, 21, 52, 53]. In the same line, we have also previously shown that compliance in meditation training in novices was associated with increased power in the lower alpha band (7–10 Hz) [54]. Since individual alpha peak is positively correlated to arousal levels [55], reduced individual alpha frequency during meditation after mindfulness training is likely to reflect greater levels of relaxation. This idea would be in line with previous studies that reported mindfulness-related decreases in arousal levels [56–59]. Reduced arousal levels during meditation could be part of the mechanisms of action behind the positive effects of mindfulness training. In this way, it is possible that a low-arousal state during meditation would reduce the emotional reactivity to self-generated thoughts (which are likely to contain worry and rumination [12] thereby facilitating emotional regulation and ultimately, mental health.

### Limitations

The main limitation of the study is the relatively small sample size (N = 41). This was due to dropouts and difficulties in data collection due to the COVID-19 pandemic. Hence, a bigger sample size is needed to confirm the here reported results. In addition, the sample consisted on a young student population with high levels of perceived stress and therefore, future studies would have to determine whether the effects of mindfulness training on EEG and mind wandering are similar in other populations. Finally, we cannot completely rule out the possibility the participants in the control group practised some sort of meditation technique during the waitlist period. This might explain why both groups showed reduced drowsiness in the second meditation session.

## Summary and conclusion

In summary, our results showed that mind wandering during meditation and their EEG correlate (occurrence of spatially widespread low-theta oscillations) are not significantly reduced after 8-weeks of mindfulness training. Nonetheless, we identified changes in EEG during meditation after mindfulness training that are consistent with what has been previously observed in highly experienced meditators. Specifically, mindfulness training was associated to a significant slowing of alpha oscillations during meditation, which is considered an EEG marker of reduced arousal levels. Our main limitation is the relatively small sample size (N = 41), which only contained young adults. Future studies using larger (and more diverse) samples in combination with more complex experience sampling paradigms are needed to identify the neuro-phenomenological changes associated to mindfulness training and their relation to its putative health benefits.

## Author Contributions

**Conceptualization:** Julio Rodriguez-Larios, Kian Foong Wong, Julian Lim.

**Data curation:** Kian Foong Wong.

**Formal analysis:** Julio Rodriguez-Larios.

**Funding acquisition:** Julian Lim.

**Investigation:** Julio Rodriguez-Larios.

**Methodology:** Julio Rodriguez-Larios.

**Project administration:** Kian Foong Wong.

**Resources:** Julian Lim.

**Software:** Julio Rodriguez-Larios, Kian Foong Wong.

**Supervision:** Julian Lim.

**Writing – original draft:** Julio Rodriguez-Larios.

**Writing – review & editing:** Julio Rodriguez-Larios, Kian Foong Wong, Julian Lim.

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
