## [Decision Letter · Decision Letter 0]

19 Mar 2024

PONE-D-24-04082Assessing the effects of an 8-week mindfulness training program on neural oscillations and self-reports during meditation practicePLOS ONE

Dear Dr. Rodriguez Larios,

Thank you for submitting your manuscript to PLOS ONE. After careful consideration, we feel that it has merit but does not fully meet PLOS ONE’s publication criteria as it currently stands. Therefore, we invite you to submit a revised version of the manuscript that addresses the points raised during the review process.

We look forward to receiving your revised manuscript.

Kind regards,

Manob Jyoti Saikia, Ph.D.

Academic Editor

PLOS ONE

Journal Requirements:

   "This study was funded by the start-up funding from the National University of Singapore to Julian Lim."

Reviewers' comments:

Reviewer's Responses to Questions

**Comments to the Author**

1. Is the manuscript technically sound, and do the data support the conclusions?

Reviewer #1: Yes

Reviewer #2: Yes

2. Has the statistical analysis been performed appropriately and rigorously? 

Reviewer #1: Yes

Reviewer #2: Yes

3. Have the authors made all data underlying the findings in their manuscript fully available?

Reviewer #1: Yes

Reviewer #2: Yes

4. Is the manuscript presented in an intelligible fashion and written in standard English?

Reviewer #1: Yes

Reviewer #2: Yes

5. Review Comments to the Author

Reviewer #1: Methods, Participants:

It says 35 participants were allocated to treatment and 24 to Waitlist. But the reported mindfulness training (N=21) and the waitlist/control was (N=20). Could you briefly explain the discrepancy?

Table 1:

Can you include any explanation on why the there was a similar decrease from Pre to Post in both Treatment and Control groups?

On alpha slowing during meditation practice:

(relative to control condition), what is the lower alpha band of the control group?

Reviewer #2: Abstract:

• Abstracts should be written in the third person rather than the first person.

Introduction Section:

• Background information on EEG methodology regarding neural correlates of mindfulness meditation should be elaborated.

• Discussion on the mechanisms behind mindfulness should be broadened, especially regarding mind wandering.

Method Section:

• Inclusion and exclusion criteria should be mentioned

• Details of sample allocation should be described.

• More demographic data of the participants are needed.

• The rationale for the unequal distribution of samples between the waitlist and treatment group should be mentioned.

Discussion section:

• The discussion does not extensively address the methodological limitations of the study, such as the small sample size and the use of a single university student population

• The discussion primarily focuses on interpreting the study's findings within the context of existing literature. However, it does not consider alternative explanations or potential confounding factors that could influence the observed results.

• Limitations of the methodology should be added.

Conclusion section:

This section should include suggestions for further research or a call to action

The entire article should be written in the third person rather than the first person.

6. PLOS authors have the option to publish the peer review history of their article (what does this mean?). If published, this will include your full peer review and any attached files.

Reviewer #1: No

Reviewer #2: No

---

## [Author Response · Author response to Decision Letter 0]

3 Apr 2024

We would like to thank reviewers for their constructive feedback. We believe that these comments have helped us to improve our manuscript significantly.

Reviewer #1

Methods, Participants:

It says 35 participants were allocated to treatment and 24 to Waitlist. But the reported mindfulness training (N=21) and the waitlist/control was (N=20). Could you briefly explain the discrepancy?

We now explain this discrepancy in the Participants subsection of the Methods. 

35 participants were allocated to treatment (i.e. mindfulness training) and 24 to waitlist. EEG was recorded before and after 8 weeks in 21 treatment and 20 control participants. The discrepancy between recruited participants and the actual sample size was due to dropouts. In this regard, note that the study was run during the COVID-19 pandemic, which significantly hindered assistance to the programme and data collection sessions. 

Table 1:

Can you include any explanation on why the there was a similar decrease from Pre to Post in both Treatment and Control groups?

Note that the pre-post changes across groups was significant for drowsiness but not for mind wandering. We now clarify this in the results section. 

Contrary to our hypothesis, Repeated Measures ANOVA did not reveal a significant Group*Session interaction in the mind wandering - focused scale (F(1,40) = 0.15, p = 0.69) nor in the alert-drowsy scale (F(1,40) = 0.05, p = 0.94). The mean scores (and their standard deviation) for each scale per group and session are depicted in Table 1. In this regard, both treatment and control groups show a similar decrease in mind wandering and drowsiness in the second session. This decrease was significant for drowsiness (F(1,40) = 7.50, p = 0.009) but not for mind wandering (F(1,40) = 0.3, p = 0.58).

Although it is difficult to know why this is the case, one possibility is that the control group practised some sort of meditation technique during the waitlist period. We now add this speculation in the Discussion. 

Finally, we cannot completely rule out the possibility the participants in the control group practised some sort of meditation technique during the waitlist period. This might explain why both groups showed reduced drowsiness in the second meditation session. 

On alpha slowing during meditation practice:

(relative to control condition), what is the lower alpha band of the control group?

I am not sure I understand this question. If you refer to the comparison between meditation and a control condition in the control group, that would be beyond the scope of this paper. In this study we wanted to assess whether mind wandering during meditation (and their putative EEG correlates) are modulated after mindfulness training. In previous studies we have assessed the EEG correlates of meditation (relative to a control condition) in both experienced practitioners and meditation-naïve controls (see Rodriguez-Larios et al., 2021).

Reviewer #2

Abstract:

• Abstracts should be written in the third person rather than the first person.

We prefer to write papers in first person to facilitate readability. 

Introduction Section:

• Background information on EEG methodology regarding neural correlates of mindfulness meditation should be elaborated.

• Discussion on the mechanisms behind mindfulness should be broadened, especially regarding mind wandering.

We modified our introduction in order to provide more background information on EEG, the neural correlates of mindfulness and mind wandering as a possible mechanism of action. 

Introduction

Mindfulness is a type of meditation practice that consists of paying attention to the present moment non-judgmentally [1]. This is usually cultivated through focused meditation, in which a particular object (e.g. the breath) is chosen as the target of attention [2,3]. The practice of mindfulness has become popular in western cultures in the last years due to its putative health benefits [4]. The mechanisms of action behind mindfulness are still debated and they are likely to involve a wide variety of factors [5,6]. 

It has been proposed that some of the positive effects of mindfulness on mental health could be mediated by reductions in mind wandering [7], which can be defined as the emergence of spontaneous, self-generated thoughts that often entail memories, future plans or fantasies [8]. In support of this idea, previous studies have shown that mindfulness trait is negatively correlated to self-reported mind wandering [9] and that mindfulness training reduces mind wandering during different cognitive tasks [10,11]. In addition, excessive mind wandering has been associated to poor mental health [12], probably due to its link to rumination and worry [13–15]. 

The neural correlates of meditation and mind wandering have been investigated through Electroencephalography (EEG). EEG is a non-invasive method that allows to record synchronized activity of large populations of neurons that are arranged orthogonal to the scalp [16]. The EEG signal is dominated by oscillatory electrical activity that is normally referred as neural oscillations. Neural oscillations have been classified according to their peak frequency (e.g. alpha = 8 -13 Hz; theta = 4 – 8 Hz, etc.) and their occurrence has been associated to different cognitive functions and mental states [17,18]. 

The EEG correlates of meditation depend on the type of meditation practice, the level of expertise of the participants and the ‘control’ condition used as a baseline [19]. In this way, it has been shown that breath focus meditation (relative to rest) is associated to decreases in alpha/beta amplitude and individual alpha frequency in experienced practitioners [20–22]. Because alpha power has been positively associated to mind wandering [23], its decrease during meditation (relative to rest) is thought to reflect reduced mind wandering [20]. This idea is further supported by studies demonstrating significantly reduced mind wandering during meditation in experienced meditators relative to novices [20,24]. 

In this study, we assessed whether mind wandering during meditation (and their associated neural correlates) change significantly in novices after 8 weeks of mindfulness training. For that purpose, we recorded EEG during meditation before and after an 8-week mindfulness training (21 active and 20 waitlist controls). Using an experience sampling paradigm, we prompted participants during meditation practice to report their level of mind wandering and drowsiness. Our (pre-registered) hypothesis was that mind wandering during meditation practice would be reduced after mindfulness training and that this would be reflected in EEG neural oscillations. To test this hypothesis, we first characterized EEG modulations associated with mind wandering during meditation. Then, we assessed whether mind wandering and/or the EEG correlates of meditation practice changed significantly after mindfulness training. 

Method Section:

• Inclusion and exclusion criteria should be mentioned

• Details of sample allocation should be described.

• More demographic data of the participants are needed.

• The rationale for the unequal distribution of samples between the waitlist and treatment group should be mentioned.

We added more information in the methods section about participants and their recruitment. 

Participants

48 participants (28 females) were recruited for the study. To be eligible for the study, participants had to be a National University of Singapore student between 21 and 35 years old. In addition, participants had to report moderate to high levels of perceived stress (Perceived Stress Scale score > 14), be willing to participate in an 8-weeks mindfulness course and be meditation-naïve. Exclusion criteria were: i) chronic physical or psychiatric illness, including all major Axis I and II disorders, ii) history of drug or alcohol abuse, and iii) long-term medication use. 35 participants were allocated to treatment (i.e. mindfulness training) and 24 to waitlist. The average age was 23.81 (SD = 2.59). EEG was recorded before and after 8 weeks in 21 treatment and 20 control participants. The discrepancy between recruited participants and the actual sample size was due to dropouts. In this regard, note that the study was run during the COVID-19 pandemic, which significantly hindered assistance to the programme and data collection sessions. 

Discussion section:

• The discussion does not extensively address the methodological limitations of the study, such as the small sample size and the use of a single university student population

• The discussion primarily focuses on interpreting the study's findings within the context of existing literature. However, it does not consider alternative explanations or potential confounding factors that could influence the observed results.

• Limitations of the methodology should be added.

We added a Limitations subsection in the Discussion where we adress some methodological limitations and possible confounders. 

Limitations

The main limitation of the study is the relatively small sample size (N=41). This was due to dropouts and difficulties in data collection due to the COVID-19 pandemic. Hence, a bigger sample size is needed to confirm the here reported results. In addition, the sample consisted on a young student population with high levels of perceived stress and therefore, future studies would have to determine whether the effects of mindfulness training on EEG and mind wandering are similar in other populations. Finally, we cannot completely rule out the possibility the participants in the control group practised some sort of meditation technique during the waitlist period. This might explain why both groups showed reduced drowsiness in the second meditation session.

Conclusion section:

This section should include suggestions for further research or a call to action

The entire article should be written in the third person rather than the first person.

Suggestions for further research are now added in the conclusion (see below). As previously mentioned, we prefer to write the article in first person to facilitate readability.

Summary and conclusion

In summary, our results showed that mind wandering during meditation and their EEG correlate (occurrence of spatially widespread low-theta oscillations) are not significantly reduced after 8-weeks of mindfulness training. Nonetheless, we identified changes in EEG during meditation after mindfulness training that are consistent with what has been previously observed in highly experienced meditators. Specifically, mindfulness training was associated to a significant slowing of alpha oscillations during meditation, which is considered an EEG marker of reduced arousal levels. Our main limitation is the relatively small sample size (N = 41), which only contained young adults. Future studies using larger (and more diverse) samples in combination with more complex experience sampling paradigms are needed to identify the neurophenomenological changes associated to mindfulness training and their relation to its putative health benefits.

---

## [Decision Letter · Decision Letter 1]

23 May 2024

Assessing the effects of an 8-week mindfulness training program on neural oscillations and self-reports during meditation practice

PONE-D-24-04082R1

Dear Dr. Rodriguez-Larios,

We’re pleased to inform you that your manuscript has been judged scientifically suitable for publication and will be formally accepted for publication once it meets all outstanding technical requirements.

Kind regards,

Manob Jyoti Saikia, Ph.D.

Academic Editor

PLOS ONE

---

## [Editor Report · Acceptance letter]

27 May 2024

PONE-D-24-04082R1 

PLOS ONE

Dear Dr. Rodriguez-Larios, 

I'm pleased to inform you that your manuscript has been deemed suitable for publication in PLOS ONE. Congratulations! Your manuscript is now being handed over to our production team.

Kind regards, 

on behalf of

Dr. Manob Jyoti Saikia 

Academic Editor

PLOS ONE